# The Biochemical Activities of the *Saccharomyces cerevisiae* Pif1 Helicase Are Regulated by Its N-Terminal Domain

**DOI:** 10.3390/genes10060411

**Published:** 2019-05-28

**Authors:** David G. Nickens, Christopher W. Sausen, Matthew L. Bochman

**Affiliations:** Molecular and Cellular Biochemistry Department, Indiana University, Bloomington, IN 47405, USA; dnickens@indiana.edu (D.G.N.); csausen@indiana.edu (C.W.S.)

**Keywords:** DNA helicase, *Saccharomyces cerevisiae*, Pif1, telomerase, telomere

## Abstract

Pif1 family helicases represent a highly conserved class of enzymes involved in multiple aspects of genome maintenance. Many Pif1 helicases are multi-domain proteins, but the functions of their non-helicase domains are poorly understood. Here, we characterized how the N-terminal domain (NTD) of the *Saccharomyces cerevisiae* Pif1 helicase affects its functions both in vivo and in vitro. Removal of the Pif1 NTD alleviated the toxicity associated with Pif1 overexpression in yeast. Biochemically, the N-terminally truncated Pif1 (Pif1ΔN) retained in vitro DNA binding, DNA unwinding, and telomerase regulation activities, but these activities differed markedly from those displayed by full-length recombinant Pif1. However, Pif1ΔN was still able to synergize with the Hrq1 helicase to inhibit telomerase activity in vitro, similar to full-length Pif1. These data impact our understanding of Pif1 helicase evolution and the roles of these enzymes in the maintenance of genome integrity.

## 1. Introduction

DNA helicases are enzymes that couple DNA binding and ATP hydrolysis to unwind double-stranded DNA (dsDNA) into its component single strands [1]. This activity is vital to many processes involved in the maintenance of genome integrity, including DNA replication, recombination and repair, transcription, and telomere maintenance [2]. As such, the genomes of all organisms encode many helicases to fulfill these roles, from approximately 10 in typical prokaryotes to 100 in eukaryotes [3]. These enzymes can be grouped into superfamilies based on the primary sequence, and structural and functional homology of their helicase domains [4].

However, many DNA helicases are multi-domain enzymes with additional activities conferred by functional modules other than their helicase motors. For instance, some helicases in the RecQ family also contain an exonuclease domain (e.g., the Werner syndrome helicase WRN [5]), a DNA strand exchange domain (e.g., RecQ-like helicase 4, RECQL4 [6]), and/or domains involved in DNA and protein interactions (RecQ C-terminal domain (RQC) and helicase and RNaseD C-terminal domain (HRDC), respectively) [7]. These accessory domains may be used in tandem with helicase activity to ensure genome integrity, or they may be functionally separable. The latter is the case with RECQL4, which has an N-terminal domain (NTD) necessary for the initiation of DNA replication with no dependence on helicase activity [8].

Similar to RecQ helicases, Pif1 family helicases are also often multi-domain proteins, especially those found in eukaryotes [9,10]. They tend to have a central helicase domain with NTDs and C-terminal domains (CTDs) that significantly vary in length and sequence conservation, even among closely related homologs. Further, with very few exceptions, these accessory domains have no identified or predicted function. Among the two *Saccharomyces cerevisiae* Pif1 family helicases, Rrm3 and Pif1, it is known that the NTD of Rrm3 is essential for its functions in vivo because truncation of the Rrm3 NTD phenocopies the null allele [11]. However, the Rrm3 NTD can be fused to the *S. cerevisiae* Pif1 helicase domain or the helicase domain of bacterial Pif1 proteins, and these chimeras can rescue the synthetic lethality of cells lacking Rrm3 and, for instance, the gene encoding the Srs2 helicase [12]. Thus, in this experimental set up, the Pif1 helicase domains are generic and interchangeable motor modules, while the Rrm3 NTD is vital for genome integrity.

We recently characterized the Pif1 helicase from the thermophilic bacterium *Thermotoga elfii* (TePif1) and investigated the function of its C-terminal WYL domain [12]. We found that the WYL domain contains a single-stranded DNA (ssDNA) binding motif that regulates and couples the ATPase and helicase activities of the enzyme. Here, we sought to extend this line of investigation to the NTD of the *S. cerevisiae* Pif1 helicase, which currently has an unknown function. Using both in vivo and in vitro approaches, we found that the Pif1 NTD is involved in the toxicity of Pif1 overexpression and impacts the ability of Pif1 to regulate telomerase activity. These data and similar investigations will further illuminate the conserved and divergent functions of Pif1 family helicases across species, leading to a better understanding of the roles of these important enzymes in genome integrity.

## 2. Materials and Methods

### 2.1. Yeast Strains, Media, and Other Reagents

*Saccharomyces *cerevisiae** strain JBBY26, a derivative of BCY123 (*MATa can1 ade2 trp1 ura3-52 his3 leu2-3*, *112 pep4::HIS3 prb1::LEU2 bar1::HISG*, *lys2::pGAL1*/*10-GAL4*) [13], harbors a dual overexpression plasmid for *TLC1* and *EST2.* The other yeast strains used in this work are derived from W303 (*MATa leu2-3,112 trp1-1 can1-100 ura3-1 ade2-1 his3-11,15*)**, which was a gift from Peter Walter (University of California, San Francisco, CA, USA). *Escherichia coli* strain Rosetta 2 (DE3) pLysS (MilliporeSigma, Burlington, MA, USA) was used for the overexpression of SUMO-tagged Pif1, Pif1ΔN, and SUMO protease (see Table 1 for expression plasmid data). For propagation and Est2/*TLC1* overproduction, JBBY26 cultures were grown in SC-Ura drop-out media. Rosetta cells were maintained on lysogeny broth (LB) medium supplemented with 50 μg/mL kanamycin and 34 μg/mL chloramphenicol. Liquid cultures were grown in 2 × YT medium (1.6% *w/v* tryptone, 1% *w/v* yeast extract, 0.5% NaCl, pH 7.0) for protein overproduction and supplemented with the same antibiotics. Radiolabeled [α-^32^P]TTP and [γ-^32^P]ATP were purchased from PerkinElmer Life Sciences. All dNTPs were purchased from New England Biolabs (Ipswich, MA, USA). Oligonucleotides were purchased from IDT (Coralville, IA, USA), and the Tel15, Tel30, and Tel50 primers used for quantitative telomerase assays were PAGE-purified. The sequences of all of the oligonucleotides used are listed in Table 2. Chemical reagents were purchased from Sigma or DOT Scientific (Burton, MI, USA). All recombinant proteins were purified as described in [14].

### 2.2. Recombinant Proteins

Plasmid pSUMO-Pif1 (Table 1) was used for the overexpression of SUMO-tagged Pif1 and was a gift from Kevin Raney. This plasmid was also used as the template to create pSUMO-Pif1ΔN to overexpress an N-terminal truncation of Pif1 lacking the first 233 amino acids of the helicase (Pif1ΔN) in *E. coli.* The overexpression and purification of these proteins proceeded as previously described for full-length Pif1 [14]. Recombinant Hrq1 was generated using baculovirus-infected insect cells as previously described [6,18]. All recombinant helicase preparations were tested for ATPase activity and the absence of contaminating nuclease activity before use in biochemical assays.

### 2.3. Telomerase Assays

In vitro telomerase assays were performed as described in [14] using telomerase-enriched extracts from *S. cerevisiae* prepared by DEAE fractionation of clarified lysates [19,20]. Each telomerase preparation was titrated to standardize activity levels before use in experiments. Reaction products were separated on 16% 19:1 acrylamide:bis-acrylamide gels containing 6 M urea. The gels were run at 2500 V for 120 min, dried, and imaged and quantified using a Typhoon 9500 scanner with ImageQuant software. Total activity was measured by quantifying the densitometry for each telomerase extension product band on a gel using ImageQuant. The sum of the measured densitometry values in a lane was reported as the total activity. Bands were corrected for the number of dT residues (i.e., the amount of α-^32^P-dTTP incorporation) and normalized to a loading control to generate corrected pixel values.

### 2.4. Southern Blotting

Wild-type cells were transformed with plasmid pRS414 (empty vector), pMB282 (Pif1), or pMB327 (Pif1ΔN) (Table 1) by the lithium acetate method [21] and selected for on media lacking tryptophan. Three transformants from each reaction were then serially restreaked for ~50 generations prior to genomic DNA (gDNA) isolation using lithium acetate and SDS [22]. Southern blots to analyze telomere length (telomere blots) were performed essentially as described [23]. Briefly, gDNA was digested overnight with *Pst*I and *Xho*I at 37 °C. Digested DNA was separated on 1% agarose gels and blotted onto Hybond-XL membranes (GE Life Sciences). The blots were probed with an *Eco*RI restriction fragment rich in C_1-3_A/TG_1-3_ sequence from plasmid pUC19(+)TEL, a gift from Katherine Friedman.

### 2.5. Helicase Assays

Fork substrates for helicase assays were constructed by incubating two partially complementary oligonucleotides (both at 1 μM) overnight at 37 °C for annealing. In each case, the helicase loading strand was end-labeled with [γ-^32^P]ATP by T4 PNK (New England Biolabs, Ipswich, MA, USA) using the manufacturer’s instructions. Labeled DNA was separated from unincorporated ^32^P using G-50 micro-columns (GE Healthcare, Chicago, IL, USA) and added to unlabeled partially complementary oligonucleotides in an annealing buffer (20 mM Tris–HCl, pH 8.0, 4% glycerol, 0.1 mM EDTA, 40 μg/mL BSA, and 10 mM DTT) [18]. The DNA–DNA fork and its RNA–DNA fork analog were prepared by heating partially complementary oligonucleotides (MB1572/MB1596 and MB1571/MB1572, respectively) (Table 2) to 75 °C, followed by slow cooling for ~2 h to room temperature. RNase inhibitors (NEB) were used during the preparation of RNA-containing substrates and during helicase assays with RNA/DNA hybrid fork substrates. All reagents were prepared with diethyl pyrocarbonate (DEPC)-treated water. Helicase reactions were performed at 30 °C for 30 min in 1× binding buffer (25 mM HEPES (pH 8.0), 5% glycerol (*w*/*v*), 50 mM NaOAc, 150 μM NaCl, 7.5 mM MgCl_2_, and 0.01% Tween-20 (*w*/*v*)) supplemented with 5 mM ATP. Reactions were stopped by the addition of 5× dye-free load buffer (50 mM Tris, pH 8.0, and 25% *w/v* glycerol) and placed on ice. Labeled fork substrates were added to a final concentration of 0.2 nM. Helicase reaction products were separated on native 12% 19:1 acrylamide:bis-acrylamide gels supplemented with 10 mM MgOAc and 5% glycerol. The Tris-Borate-EDTA running buffer (45 mM Tris-borate and 1 mM EDTA, pH 8.0) was supplemented with 2.5 mM MgOAc. Gels were run at 100 V for 30–45 min, dried, and imaged and quantified using a Typhoon 9500 scanner with ImageQuant software.

### 2.6. Electrophoretic Mobility Shift Assays

Substrates for electrophoretic mobility shift assays (EMSAs) were prepared by end-labeling oligonucleotides. The Tel15 and Tel30 substrates were used in ssDNA binding reactions, and forked dsDNA substrates were prepared as described above for helicase assays. All “Tel” oligonucleotides contained the *S. cerevisiae* telomere repeat sequence TG_1-3_. Oligonucleotides were labeled with T4 polynucleotide kinase and [γ-^32^P]ATP under standard conditions. Labeled oligonucleotides were separated from the unincorporated label using G50 micro-columns. Binding reactions were performed in 1× binding buffer. Radiolabeled substrates were boiled, placed on ice, and added to binding reactions to a final concentration 0.2 nM. When present, ATP was added to binding reactions to a final concentration of 4 mM. Binding reactions were incubated at 30 °C for 30 min and mixed with 5× dye-free loading buffer (50 mM Tris (pH 8.0) and 25% glycerol (*w*/*v*)). The reactions were separated on native 4% 37.5:1 acrylamide:bis-acrylamide gels in 1× Tris-glycine running buffer (25 mM Tris (pH 8.0) and 185 mM glycine, pH 8.8). Gels were run at 100 V for 30–45 min, dried, and imaged and quantified using a Typhoon 9500 scanner with ImageQuant software. All data were plotted and, where appropriate, fit with curves using GraphPad software.

### 2.7. Statistical Analysis

All data were analyzed and graphed using GraphPad Prism 6 software. The reported values are averages of ≥3 independent experiments, and the error bars are the standard deviation. *P*-values were calculated as described in the figure legends, and we defined statistical significance as *p* < 0.01.

## 3. Results

### 3.1. Removal of the Pif1 NTD Relieves the Toxicity of Pif1 Overexpression

It has previously been reported that the overexpression of Pif1 is toxic to *S. cerevisiae* [24]. This was also the case in our strain background (Pif1 glucose vs. Pif1 galactose, *p* < 2.2 × 10^−6^; Figure 1A), and we wanted to test whether the Pif1 NTD is involved in this effect. To investigate this question, we overexpressed a Pif1 N-terminal deletion construct lacking 233 amino acids of the helicase (Pif1ΔN). Although cell growth did not completely recover to levels observed in the absence of Pif1 overexpression (Pif1 glucose vs. Pif1ΔN galactose, *p* = 0.0002), we still found that the toxicity of overexpression was significantly relieved in the absence of the Pif1 NTD (Pif1 galactose vs. Pif1ΔN galactose, *p* < 1.2 × 10^−6^; Figure 1A). Pif1′s large NTD is of an unknown function, both in vivo and in vitro. It is predicted to be natively disordered, with the exception of five putative short α helices (Figure 1B,C). Further, Pif1 is a multi-functional helicase [9], and truncation of its NTD may impact one or more of its in vivo roles. Since the Pif1 helicase has been firmly established as a regulator of telomerase activity and a suite of assays already exist to probe this function, we next decided to determine how Pif1ΔN behaved in this role.

### 3.2. In Vitro Pif1ΔN Is a More Potent Telomerase Inhibitor Than Full-Length Pif1

We recently found that Pif1 impacts in vitro telomerase activity in a biphasic manner: low concentrations of Pif1 increase telomerase activity, but high Pif1 concentrations decrease telomerase activity [14]. Using a 15-nt telomerase repeat sequence (TG_1-3_) oligonucleotide substrate (Tel15), an *S. cerevisiae* extract enriched for telomerase activity, and recombinant Pif1, we demonstrated a similar phenomenon here in Figure 2A,B. The addition of 25–125 nM Pif1 significantly (*p* < 0.01) increased both Type I processivity (i.e., direct extension of Tel15 by 1–7 nt to generate products T16–T22) and Type II processivity (i.e., further elongation after Type I processivity to generate products T23–T34) [26] by telomerase (Figure 2B). However, with increasing Pif1 concentration from 25–125 nM, this stimulation of telomerase activity decreased in a dose-dependent manner for Type I processivity products. At 250 nM Pif1, total telomerase activity was further decreased to levels indistinguishable from reactions lacking Pif1 (Figure 2A,B), and additional increases in Pif1 concentration up to 360 nM significantly decreased telomerase activity relative to reactions lacking Pif1 [14].

We next performed similar in vitro telomerase reactions containing Pif1ΔN. Compared to full-length Pif1, the effect of Pif1ΔN on telomerase activity was quite different (Figure 2). In this case, Pif1ΔN only stimulated telomerase activity at the lowest helicase concentration tested (25 nM), and this stimulation was only significant (*p* < 0.0001) for Type I processivity (Figure 2C,D). At higher concentrations of Pif1ΔN, the helicase significantly inhibited both Type I and especially Type II processivity by ≥50% relative to reactions lacking added helicase. Thus, Pif1ΔN was a more effective inhibitor of in vitro telomerase activity than full-length Pif1.

### 3.3. Chronic Overexpression of Pif1ΔN Does Not Lead to Telomere Length Crisis

In the absence of telomerase activity, telomere length decreases during every cell generation until the telomeres reach a critically short point known as crisis [27]. When crisis occurs, most cells senesce. Another way to shorten telomeres is to overexpress Pif1, which decreases telomere length in a dose-dependent manner [24,28]. Our in vitro data above indicate that Pif1ΔN is a more potent inhibitor of telomerase activity than the full-length Pif1 helicase, and thus, the expression of Pif1ΔN in vivo may lead to rapid shortening of telomeres and crisis even in the presence of functional telomerase. To investigate this hypothesis, we chronically overexpressed Pif1 and Pif1ΔN by galactose induction from a multi-copy vector in wild-type yeast (Appendix A).

In the absence of induction, cultures of cells carrying either the Pif1- or Pif1ΔN-encoding vector grew just as well as cultures harboring an empty vector over three successive restreaks (~75 generations [29]) (Figure 3). In the presence of galactose, cultures of Pif1 overexpressing cells grew more slowly than cells containing the Pif1ΔN or empty vector, as expected based on our results in Figure 1A. However, upon serial restreaking of the cells overexpressing Pif1, they ceased to grow by streak three, suggesting that cells had undergone telomere length crisis between 50 and 75 generations (Figure 3). In contrast, cultures of cells overexpressing Pif1ΔN displayed no gross growth defect relative to cells containing the empty vector. Thus, as opposed to the in vitro data (Figure 2), these data indicate that Pif1ΔN is a less potent inhibitor of telomerase activity than full-length Pif1.

### 3.4. Pif1ΔN Expression Has a Mild Effect on Bulk Telomere Length In Vivo

To directly analyze the effect of Pif1ΔN on telomere length in vivo, we expressed Pif1 or Pif1ΔN in wild-type cells for ~50 generations and performed Southern blotting to examine telomere length. Here, we did not overexpress the helicases by galactose induction because we could not recover enough cells from Pif1 overexpressing cultures after multiple restreaks. Instead, we expressed Pif1 and Pif1ΔN under the control of the *RRM3* promoter on a single-copy plasmid, conditions that mimic endogenous expression levels and can rescue *pif1* mutant phenotypes [16]. As shown in Figure 4, the overexpression of Pif1 (one copy on the chromosome, one copy on the plasmid) resulted in telomere shortening, as expected [30]. Pif1ΔN expression also caused telomere shortening, but to an intermediate length between wild-type and Pif1-overexpressing cells. This effect is discussed below in Section 4.1.

### 3.5. Truncation of the Pif1 N-Terminus Affects DNA Binding and Unwinding by the Helicase

To better understand the biochemical function of the Pif1 NTD, we compared DNA binding and unwinding by recombinant Pif1 and Pif1ΔN. Using the Tel15 substrate from the in vitro telomerase assays in EMSAs, we found that the affinity of Pif1ΔN for this ssDNA (*k_d_* = 390 ± 58 pM) was nearly 300-fold greater than full-length Pif1 (*k_d_* = 112 ± 12.7 nM) (Figure 5A). We previously found that Pif1 bound to the longer Tel30 substrate with higher affinity than Tel15 (*k_d_* = 33.5 vs. 113 nM, respectively [14]), so we also assessed binding of Pif1ΔN to Tel30. In contrast to the full-length helicase, Pif1ΔN displayed similarly tight binding to Tel30 ssDNA (*k_d_* = 410 ± 31 pM) compared to Tel15 (Appendix A), indicating that the NTD was responsible for the >3-fold increase in *k_d_* for Tel30 vs. Tel15 exhibited by Pif1 [14].

Next, we investigated DNA unwinding by Pif1 and Pif1ΔN. Both helicases displayed similar levels of activity on a forked substrate with a short dsDNA region and ssDNA telomeric repeat sequence arms (Pif1 *k*_1/2_ = 2.2 nM vs. Pif1ΔN *k*_1/2_ = 1.1 nM) (Figure 5B,C). It is known that RNA–DNA hybrids stimulate Pif1 helicase activity when the enzyme tracks along the DNA strand of the substrate due to an increase in processivity [31]. To determine if Pif1ΔN can still be stimulated by RNA–DNA hybrids, we repeated the unwinding reactions using a hybrid fork. Here, we were able to recapitulate the stimulation in activity displayed by full-length Pif1 (Figure 5B), but this effect was largely abrogated with Pif1ΔN (Figure 5C). Indeed, where Pif1 DNA unwinding was stimulated >7-fold by the RNA–DNA hybrid substrate, Pif1ΔN activity was only stimulated 2.6-fold.

To determine what may have caused this decrease in activity on RNA–DNA hybrids, we performed helicase time course assays. However, we found that both Pif1 and Pif1ΔN unwind the hybrid fork with similar kinetics (*t*_1/2_ ≈ 18 min) (Appendix A). Thus, the decreased activity of Pif1ΔN relative to Pif1 was not due to a slower rate of unwinding. Subsequently, we measured the affinity of Pif1 and Pif1ΔN for the RNA–DNA hybrid fork. As with the Tel15 ssDNA (Figure 5A), Pif1ΔN bound more tightly to the hybrid fork than Pif1 (Figure 5D) but only by ~5-fold. In this case however, both Pif1 and Pif1ΔN bound the hybrid fork substrate with greater affinity than binding to Tel15 ssDNA (Pif1 *k_d_* = 0.67 ± 0.085 nM vs. 112 ± 12.7 nM, respectively; Pif1ΔN *k_d_* = 0.135 ± 0.013 nM vs. 0.390 ± 0.058 nM, respectively). Therefore, the >160-fold increase in binding affinity for the RNA–DNA hybrid fork relative to ssDNA by Pif1 underpinned the stimulation in hybrid fork unwinding relative to the DNA–DNA fork (Figure 5B), likely due to increased processivity [31]. Pif1ΔN displayed a much more modest <3-fold increase in *k_d_* for the RNA–DNA hybrid fork relative to ssDNA and, consequently, was less stimulated by the hybrid fork in helicase assays.

### 3.6. Is the Pif1 NTD Necessary to Interact with Hrq1?

We previously demonstrated that Pif1 and the RecQ family helicase Hrq1 synergistically regulate telomerase activity both in vivo [23] and in vitro [14], and this may be due to a direct physical interaction between the helicases. As disordered regions in proteins are often sites of protein–protein interaction [32], we wondered if truncation of the disordered NTD of Pif1 would affect the synergism with Hrq1 in in vitro telomerase assays. Using a 50-nt ssDNA substrate (Tel50) to enable binding of both helicases and titrating equimolar amounts of Hrq1 and Pif1ΔN into telomerase activity reactions, we found that the Hrq1 + Pif1ΔN combination was a potent inhibitor of telomerase activity at all concentrations tested (Figure 6A,B). This is in contrast to Hrq1 alone, which mildly stimulates telomerase activity at high concentrations [14], and Pif1ΔN alone, which stimulates telomerase activity at low concentrations and inhibits it at greater concentrations (Figure 2C,D). Therefore, Hrq1 and Pif1 can still interact to regulate telomerase activity in the absence of the Pif1 NTD.

## 4. Discussion

To begin to determine the roles of the non-helicase domains of Pif1, we compared the activity of full-length Pif1 and a Pif1 truncation lacking the entirety of the natively disordered NTD (Pif1ΔN). Both in vivo and in vitro, Pif1 and Pif1ΔN displayed obvious differences in assays for cell growth, telomerase inhibition, and biochemical activities, indicating that the Pif1 NTD is critical for the proper functioning of the helicase. The implications of the results presented above are discussed below.

### 4.1. The Pif1 NTD Is Involved in Regulating Telomerase Activity

It has been shown that Pif1 induces replication stress at telomeres, with excess Pif1 activity yielding detrimental increases in telomeric ssDNA [24]. The results in Figure 1A suggest that removing the Pif1 NTD alleviates the majority of the cellular stress associated with Pif1 overexpression, indicating that Pif1ΔN may have less of an impact on telomere biology than full-length Pif1, despite increased helicase activity. Indeed, analyzing telomere length in cells expressing Pif1ΔN vs. overexpressing Pif1 demonstrated that Pif1ΔN causes mild telomere shortening relative to excess Pif1 activity (Figure 4). However, in vitro telomerase assays demonstrated that Pif1ΔN is a more active inhibitor of telomerase activity (Figure 2C,D) than Pif1 (Figure 2A,B). What explains this apparent contradiction?

We have found that the standard in vitro telomerase inhibition reaction classically used to characterize Pif1 activity [20] is actually a poor reflection of telomerase regulation in vivo [14,23]. Indeed, with the minimal system (i.e., a telomeric repeat sequence ssDNA substrate, telomerase activity-enriched cellular extract, and recombinant helicase), one fails to accurately demonstrate the activities of both the Hrq1 and Pif1 helicases. Based on in vivo data [23,30], these helicases are known telomerase inhibitors, but using them individually in reactions indicates that Hrq1 stimulates telomerase activity, while Pif1 both stimulates and inhibits telomere lengthening based on helicase concentration [14]. When combined, however, the helicases function synergistically in vitro to mimic the telomerase inhibition demonstrated in vivo. In other words, the in vitro telomerase activity assay is too biochemically reductive without supplementing the telomerase-enriched extract with additional recombinant proteins to reflect physiological conditions. That appears to be the case here with Pif1ΔN. When recombinant Hrq1 was added to telomerase activity assays along with Pif1ΔN, the combination of these helicases yielded strong telomerase activity inhibition (Figure 6A,B).

Alternatively, the differences between the in vitro and in vivo data could be due to lack of post-translational modifications (PTMs) in our recombinant Pif1 and Pif1ΔN preparations. These enzymes were prepared by overexpression in *E. coli* and thus lack eukaryotic-type PTMs. Indeed, in vivo, it is known that the activity of Pif1 is affected by PTMs [33,34,35]. For instance, phosphorylation of the C-terminus of Pif1 is necessary for the helicase to inhibit telomerase at DNA double-strand breaks [33]. Perhaps Pif1ΔN lacks an activating PTM because the modified site is missing in the truncated protein, and thus Pif1ΔN is less able to inhibit telomerase activity in vivo than full-length Pif1 (Figure 4). To address this issue, future work should investigate the spectrum of PTMs found on Pif1 vs. Pif1ΔN using mass spectrometry.

### 4.2. Is the Pif1 NTD a Site of Protein–Protein Interaction?

Since natively disordered regions of proteins are often sites of protein binding [32], we initially suspected that the Pif1 NTD may be responsible for the putative physical interaction with the Hrq1 helicase that affects telomere length homeostasis [14]. However, as stated above, recombinant Hrq1 and Pif1ΔN together still inhibited in vitro telomerase activity to a greater extent than either enzyme individually (Figure 2C,D and Figure 6A,B and [14]). This could indicate that Pif1 and Hrq1 do not need to physically interact to regulate telomerase activity but merely need to bind to the same DNA substrate. Alternatively, the Hrq1 interaction motif on Pif1 may be located outside of the NTD, and thus, the Pif1ΔN truncated protein is still able to bind to Hrq1. Further, the concentrations of recombinant protein in the telomerase assays may be artificially high enough to force an interaction between Hrq1 and Pif1ΔN that would otherwise be decreased or abolished in the absence of the Pif1 NTD.

We attempted to distinguish between these possibilities by assaying for Hrq1–Pif1 and Hrq1–Pif1ΔN interactions by co-purification of His_6_-tagged Hrq1 and untagged Pif1 or Pif1ΔN on Ni-affinity resin. These experiments failed to demonstrate an association between the helicases, so we then assayed for interaction between the helicases using the same recombinant proteins and the bifunctional protein crosslinking compound disuccinimidyl sulfoxide (DSSO). Unfortunately, we also failed to identify Hrq1–Pif1 and Hrq1–Pif1ΔN crosslinks by mass spectrometry. Crosslinking compounds vary in the chemistry of their reactive groups and linker length that the crosslinks can span [36], so a survey of additional crosslinkers will be conducted in an attempt to find in vitro conditions suitable for Hrq1–Pif1 crosslinking. Such reactions will also be performed in the presence of a ssDNA substrate and/or telomerase-enriched extract if a ternary or quaternary complex is necessary to stabilize the putative Hrq1–Pif1 interaction. Finally, we also plan to examine in vivo telomere length in wild-type and *hrq1*Δ strains expressing Pif1ΔN at Pif1-endogenous and overexpression levels to directly gauge its effects on telomerase activity.

### 4.3. Is the Pif1 NTD a Site of Nucleic Acid Interaction?

We previously demonstrated that the non-helicase CTD (a predicted WYL domain [37]) of the bacterial TePif1 helicase is an accessory of the ssDNA binding domain [12]. This WYL domain impacts both ATP hydrolysis and DNA unwinding, coupling the two to regulate the biochemistry of the helicase. Here, the Pif1 NTD was similarly found to affect DNA binding and helicase activities. However, Pif1ΔN bound both ssDNA (Figure 5A) and forked DNA (Figure 5D) substrates with a higher affinity than full-length Pif1. This suggests that the NTD inhibits ssDNA binding by the helicase. This inhibition could be due to a native disorder in the NTD that requires a binding partner to attain the active Pif1 cellular conformation and proper ssDNA binding affinity.

It should also be noted, however, that Pif1 can interact with RNA in the context of RNA–DNA hybrids, and that this interaction stimulates helicase activity by increasing Pif1 processivity [31]. Although Pif1ΔN DNA unwinding was stimulated by an RNA–DNA hybrid fork relative to activity on an analogous DNA–DNA fork (Figure 5C), the extent of this stimulation was diminished relative to full-length Pif1 (Figure 5B). Based on all of these data, we therefore hypothesize that the Pif1 NTD is involved in single-strand nucleic acid interaction, both ssDNA and ssRNA, and that these interactions also affect similar interactions of the helicase domain with single-stranded nucleic acids. Ideally, one could observe these interactions by comparing atomic-resolution structures of apo- and substrate-bound Pif1. To date, however, no high-resolution X-ray crystal or cryo-electron microscopy structures exist for a full-length multi-domain PIF1 helicase bound to a nucleic acid substrate.

### 4.4. PIF1 Perspectives

This work represents a first step toward the dissection of the functions of the non-helicase domains of *S. cerevisiae* Pif1. We found that the Pif1 NTD affects telomerase regulation and the basic biochemistry of the helicase. However, much work remains to be done, including structure–function analyses, surveying the effects of the Pif1ΔN truncation on known genetic interactions of *PIF1* (e.g., the synthetic lethality of *pif1Δ pol2-16* [38]), and determining the effects of Pif1ΔN on the myriad of other processes that Pif1 is involved in in vivo (e.g., Okazaki fragment processing [39] and break-induced repair [34]). Similar sets of assays should also be performed with Pif1ΔC constructs lacking the CTD, which is also predicted to be natively disordered (Figure 1B), as well as truncations of both the NTD and CTD to yield the helicase domain in isolation.

Additionally, these types of experiments will be necessary across the PIF1 helicase family. For instance, the second PIF1 helicase in *S. cerevisiae*, Rrm3, also has largely uncharacterized NTDs and CTDs, but Pif1 and Rrm3 perform very different functions in vivo [9]. Thus, the functions of the Pif1 NTD and/or CTD on Pif1 biology will likely not mirror those of the Rrm3 NTD and CTD on its activities. Further, these investigations will yield information about the evolutionary conservation of the functions of non-helicase domains in the Pif1 family. As many of these uncharacterized domains are predicted to be natively disordered but have divergent sequences [9,10], it will be interesting to determine if conservation of the disorder is important for function, allowing drift in the primary sequence to occur at a higher rate than in the globular helicase domain. Finally, the human Pif1 helicase (hPif1) is disease-linked when mutated [40], and understanding the roles of its non-helicase domains will shed further light on why hPif1 mutations are related to carcinogenesis and perhaps suggest treatment strategies.

## Figures and Tables

**Figure 1 genes-10-00411-f001:**
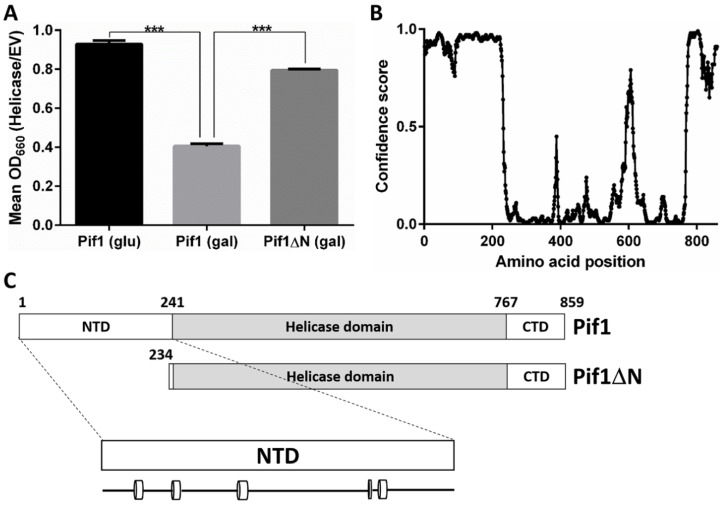
The disordered N-terminus of Pif1 is involved in Pif1 overexpression toxicity. (**A**) Removal of the Pif1 N-terminal domain (NTD) relieves the toxicity of Pif1 overexpression in *S. cerevisiae*. Galactose-inducible expression vectors for Pif1 and Pif1ΔN, as well as empty vector (EV) were transformed into wild-type cells, and the growth of these strains was monitored at an optical density of 660 nm (OD_660_) for 48 h. The mean OD_660_ during the entire time course of the cells containing the Pif1 or Pif1ΔN expression vectors was normalized to that of cells containing the empty vector to compare growth in the absence of induction (glu) and upon chronic overexpression (gal). The graphed values represent the average of ≥3 independent experiments, and the error bars correspond to the standard deviation. *** *p* < 0.0001. (**B**) The NTD of Pif1 is predicted to be natively disordered. The primary sequence of *S. cerevisiae* Pif1 was analyzed using the DISOPRED3 disorder prediction program [25], and the predicted disorder is plotted for each amino acid. The confidence score represents the likelihood that the amino acid resides in a disordered region. (**C**) Domain schematics of full-length Pif1 and Pif1ΔN. The first ATP hydrolysis motif of the Pif1 helicase domain begins at residue 241 [9], but sequence alignments, domain prediction, and disorder prediction indicate that the helicase domain itself includes residues 234–240 as well. The predicted secondary structure within the Pif1 NTD is shown below, with cylinders denoting α-helices and the lines connecting them denoting random coil.

**Figure 2 genes-10-00411-f002:**
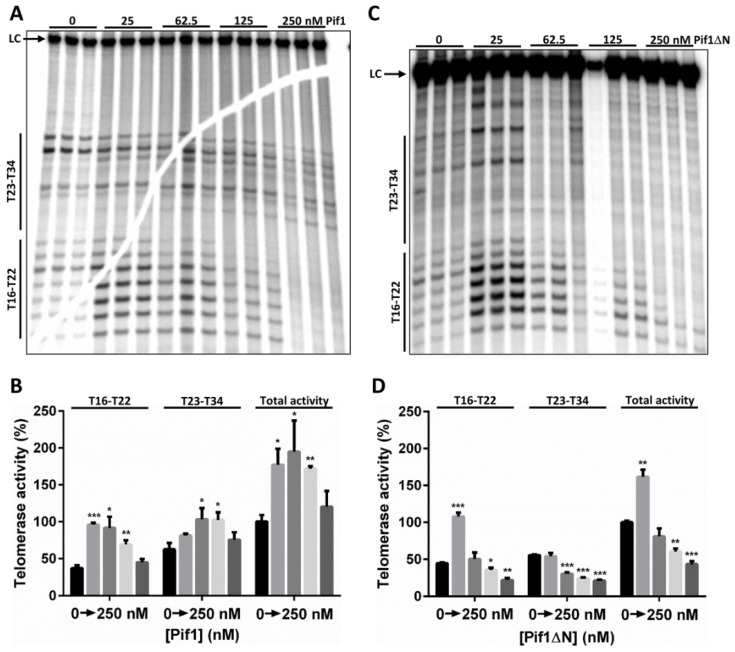
Pif1ΔN is a potent in vitro telomerase inhibitor. (**A**) In vitro telomerase extension of the Tel15 substrate in the absence or presence of the indicated concentration of recombinant Pif1. The Type I (T16–T22) and Type II (T23–T34) extension products are noted on the left. LC, loading control. (**B**) Quantification of the telomerase activity shown in (A); mean values and standard deviation are shown. Total activity refers to the total amount of signal in the lanes. (**C**) In vitro telomerase extension of the Tel15 substrate in the absence or presence of the indicated concentration of recombinant Pif1ΔN. (**D**) Quantification of the telomerase activity shown in (C). Significant differences were determined by multiple *t* tests using the Holm–Sidak method, with α = 5% and without assuming a consistent standard deviation (SD) (*n* = 3). *, *p* < 0.01; **, *p* < 0.001; and ***, *p* < 0.0001; all comparisons were made against the reactions lacking added helicase. For a graphical description of the in vitro telomerase activity, including Type I and Type II processivity and the roles of Pif1 in this process, please see [14].

**Figure 3 genes-10-00411-f003:**
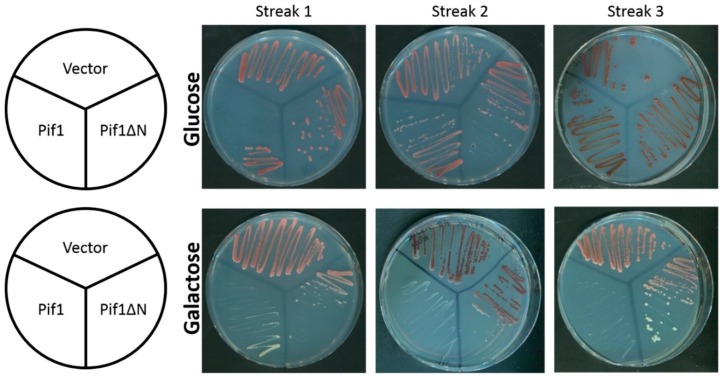
Chronic overexpression of Pif1 leads to cellular senescence. Wild-type yeast cells were transformed with a multi-copy empty vector (Vector) or vector encoding Pif1 or Pif1ΔN under the control of the inducible *GAL1*/*10* promoter. When grown on media containing glucose to inhibit overexpression, all strains produced colonies with similar growth kinetics over many successive restreaks. When grown on media containing galactose to induce overexpression of Pif1 or Pif1ΔN, the Pif1 overexpressing cells grew very slowly and failed to grow during the third restreak, indicating that these cells had senesced. Cells overexpressing Pif1ΔN, however, displayed no overt growth defect.

**Figure 4 genes-10-00411-f004:**
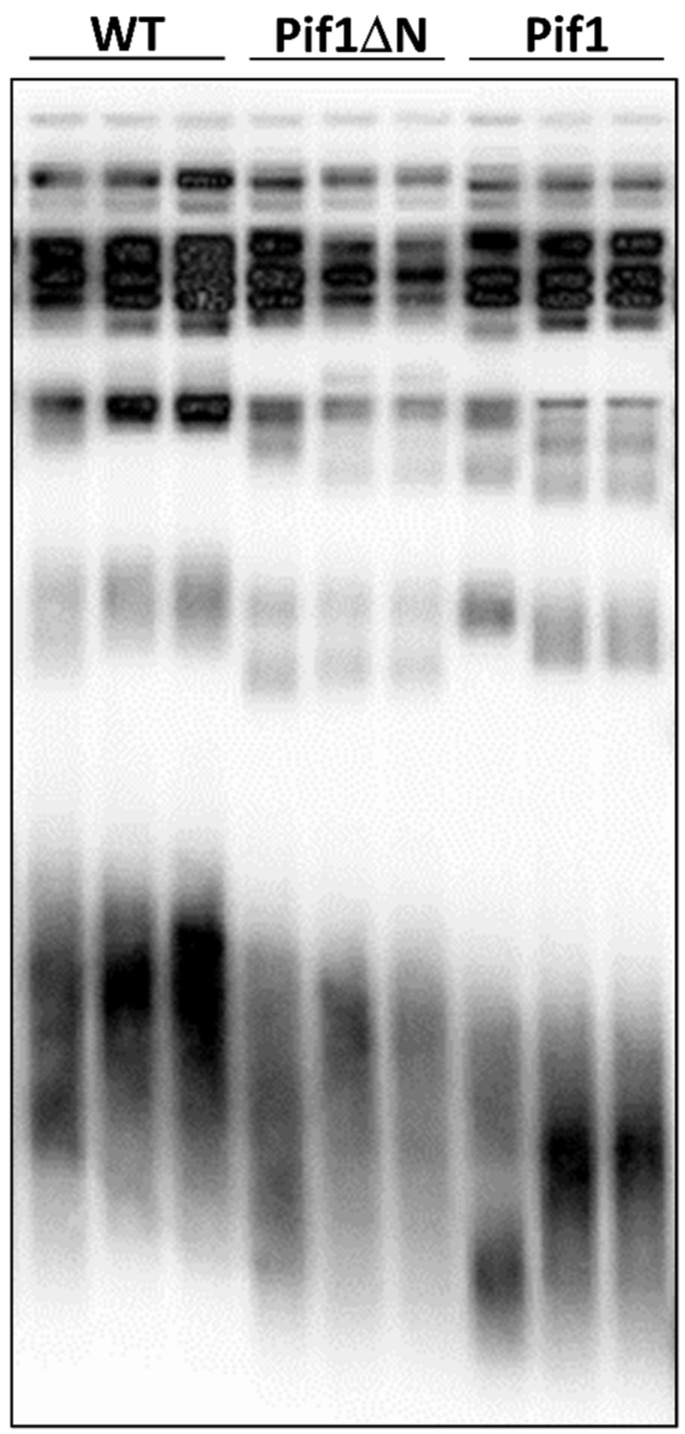
Pif1ΔN expression slightly decreases telomere length. A telomere blot of genomic DNA (gDNA) from wild-type cells that were transformed with empty vector (WT) or an expression plasmid for full-length Pif1 or Pif1ΔN is shown. Three independent clones from each plasmid transformation were examined after restreaking for ~50 generations.

**Figure 5 genes-10-00411-f005:**
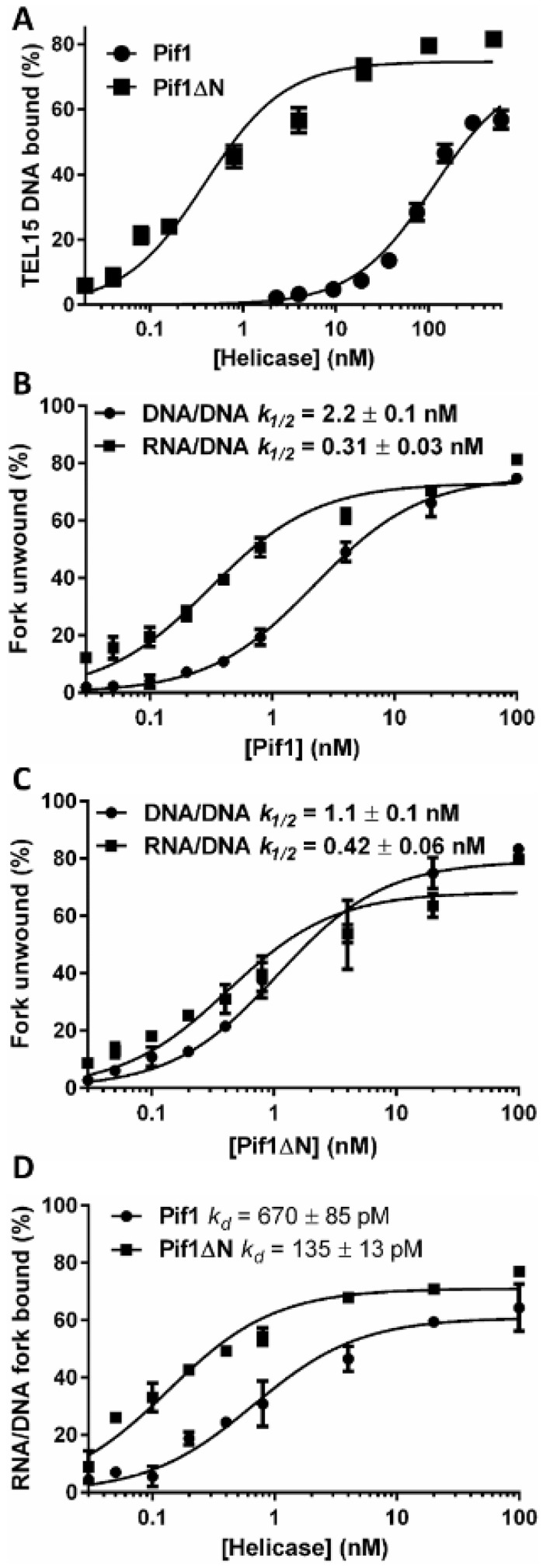
The NTD of Pif1 affects DNA binding and helicase activities. (**A**) Pif1ΔN binds the Tel15 single-stranded DNA (ssDNA) substrate much more tightly than full-length Pif1. The plotted values represent the results of EMSAs using radiolabeled Tel15 and the indicated concentrations of Pif1 and Pif1ΔN protein. (**B**) Pif1 helicase activity is stimulated by RNA–DNA hybrids. Unwinding of a radiolabeled DNA–DNA fork substrate and an analogous RNA–DNA fork were monitored with native gels at the indicated concentrations of Pif1. (**C**) Pif1ΔN displays minor helicase activity stimulation in the presence of the RNA–DNA hybrid fork. (**D**) Both Pif1 and Pif1ΔN have higher affinity for the RNA–DNA hybrid fork substrate the than Tel15 ssDNA. All data points represent the average of ≥3 independent experiments, and the error bars correspond to the standard deviation. The plotted data were fit with hyperbolic curves using GraphPad Prism software.

**Figure 6 genes-10-00411-f006:**
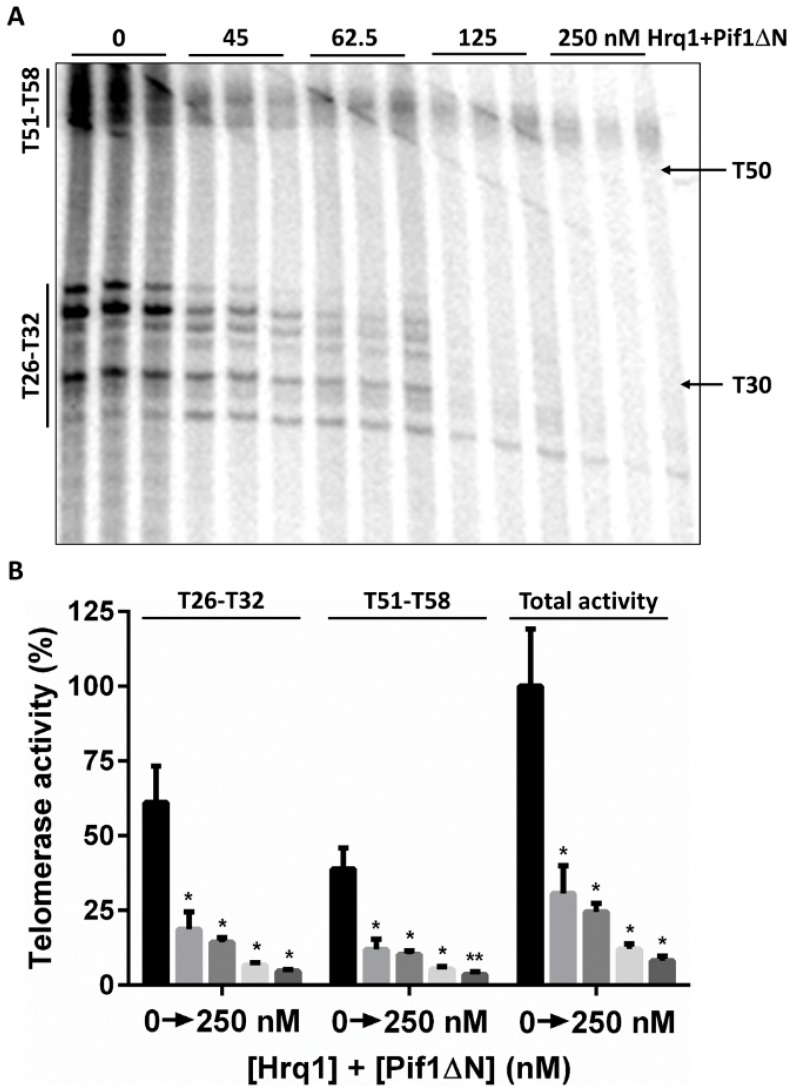
Hrq1 and Pif1ΔN are still able to interact to regulate telomerase activity. (**A**) In vitro telomerase extension of the Tel50 substrate in the absence or presence of the indicated concentration of both recombinant Hrq1 and Pif1ΔN (e.g., 45 nM indicates that 45 nM of each helicase was added). The direct extensions (Type I; T51–T58) and nuclease-cleaved extension products (T26–T32) [14] are noted on the left. The positions of the Tel30 and Tel50 markers are shown on the right. (**B**) Quantification of the telomerase activity shown in (A); mean values and standard deviation are shown. Total activity refers to the total amount of signal in the lanes. Significant differences were determined by multiple *t* tests using the Holm–Sidak method, with α = 5% and without assuming a consistent SD (*n* = 3). *, *p* < 0.01 and **, *p* < 0.001; all comparisons were made against the reactions lacking added helicase.

**Table 1 genes-10-00411-t001:** Plasmids used in this study.

Name	Description
pRS414	*TRP1*, *ARS1*, *CEN4* empty vector [15]
pMB282	Pif1-3xFLAG cloned into pRS414 under the control of the *RRM3* regulatory sequences [16]
pMB327	Pif1ΔN-3xFLAG cloned into pRS414 under the control of the *RRM3* regulatory sequences [16]
pESC-URA	Multi-copy vector enabling epitope tagging of genes cloned under the control of the bidirectional *GAL1,10* promoter
pUC19(+)TEL	pUC19 harboring yeast telomeric repeat sequence DNA cloned into the *Eco*RI site [17]
pMB526	Pif1 cloned into pESC-URA, enabling galactose induction and epitope-tagged with FLAG
pMB540	Pif1ΔN cloned into pESC-URA, enabling galactose induction and epitope-tagged with FLAG
pSUMO-Pif1	Nuclear isoform of Pif1 cloned into the pSUMO vector
pSUMO-Pif1ΔN	Pif1ΔN cloned into the pSUMO vector

**Table 2 genes-10-00411-t002:** Oligonucleotides used in this study.

Name	Sequence (5′-3′)	Purpose
MB1571	*rGrGrUrGrUrGrGrUrGrUrGrGrGrUrGrUrGrGrUrGrUrGrGrGrUrCrArCrArGrUrGrArGrUrGrUrArUrCrGrCrArArG	RNA–DNA hybrid fork substrate
MB1572	GAACGCTATGTGAGTGACACTGGGTCACCACACCCACACCACACC	Fork substrates
MB1596	GGTGTGGTGTGGGTGTGGTGTGGGTCACAGTGAGTGTATCGCAAG	DNA–DNA fork substrate
Tel15	TGTGGTGTGTGTGGG	EMSAs and in vitro telomerase assays
Tel30	CGCCATGCTGATCCGTGTGGTGTGTGTGGG	EMSAs
Tel50	GTGTGGGTGTGGTGTGGGTGTGGTGTGGGTGTGTGGTGTGGTGTGTGTGG	In vitro telomerase assays

* “r” denotes an RNA base; EMSA: electrophoretic mobility shift assay.

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
