# Peer review of "The Biochemical Activities of the Saccharomyces cerevisiae Pif1 Helicase Are Regulated by Its N-Terminal Domain"

_genes, 2019, doi:10.3390/genes10060411_

Round 1

Reviewer 1 Report

The manuscript describes the study of the function of the NTD of Pif1 in S. cerevisiae with molecular and biochemical approaches. The experiments seem to be well conducted and the conclusions are supported by the data. Although the document is very well written, there are a few language corrections to make (see below). However, a considerable part of the Discussion section is dedicated to future perspectives, which do not add to the discussion of the results. It would be advisable to decrease drastically future perspectives to focus on the discussion of the data. Given the relevance of this study to understand fundamental aspects of replication, I recommend that this manuscript should be published after correction these issues and those I present below.

Materials and Methods

Lines 71-72

There are parts of the text in italic.

Line 75

It is more correct to say cultures were grown instead of “…cells were grown…”.

Line 114

Correct “…as describe below.”.

Results

Lines 167-168

The authors’ conclusion “low concentrations of Pif1 increase telomerase activity, but high Pif1 concentrations decrease telomerase activity“ doesn’t seem to be supported by the data. In fact, telomerase activity increases with Pif1N concentrations up to 125 nM but at 250 nM the activity is similar to the assay without Pif1N. However, below, in the same paragraph, they explain correctly the conclusion based on reference 14 and data not shown. In order to make the text clearer, it would be better to take the conclusion after the presentation of the data (particularly the reference 14 and data not shown).

Fig. 2

Please describe the meaning of LC in the legend.

Line 223-224

Unless the authors mean actual growth of cells, they should say “cultures […] grew as well…” instead of “cells […] grew as well…”. There are more examples below in the manuscript, please correct.

Line 224

Please explain how did the authors concluded that three restreaks corresponds to nearly 75 generations.

Line 233

Please avoid determinism in “…biochemical purpose of the Pif1 NTD…”. Pif1 NTD does not have a purpose, it simply has a function.

Author Response

Our replies to your comments have been uploaded as a Word document (Reviewer 1.docx).

Reviewer 2 Report

Pif1 is a highly conserved protein with fundamental roles in okazaki fragment processing, DNA replication across natural pausing elements, telomere length regulation, break inducible replication and mitochondrial DNA maintenance.

The functional studies of S.cerevisiae Pif1 (scPif1) and S. pombe pfh1 (sppfh1) protein domains are important as all the cellular roles of this DNA helicase have been discovered using these two model organisms.

In this manuscript the authors address the in vivo phenotypes caused by high levels of the Pif1DN mutant protein that lacks the first N-terminal 233 amminoacids of scPif1 and its biochemical properties.

Major findings

in vivo:

1) High levels of scPif1DN do not heavily reduce growth rates or induce cellular senescence as high levels of the wild type scPif1 do.

in vitro:

1) scPif1DN mutant protein is a better inhibitor of the telomerase than scPif1 in telomerase extension assays.

2) scPif1DN binds telomeric repeats and RNA-DNA fork substrates with higher affinity than scPif1.

3) scPif1DN unwinds DNA-DNA forks with higher efficiency than scPif1.

4) scPif1DN unwinds RNA-DNA forks slightly better than scPif1.

5) scPif1DN synergizes with Hrq1 better than scPif1 in inhibiting the telomerase in telomere extension assays.

Conclusions

1) Cellular toxicity and senescence induced by high levels of scPif1 are due to its N-terminal domain.

2) The N-terminal domain of scPif1 is involved in the Pif1 mediated down-regulation of the telomerase.

3) Surprisingly, although scPif1DN is a better inhibitor of the telomerase in vitro, high levels of it induce much less cytotoxicity compared to high levels of scPif1.

The manuscript deserves publication in Genes because it adds new pieces of knowledge to the function of the unstructured N-terminal domain of scPif1.

If it does not take long time and if the experimental approaches are feasible, I suggest addressing the two following major points before publication (not mandatory). 

Major Points

1) It looks like that the authors never checked the cellular protein levels of scPif1 and scPif1DN under their over-expressing conditions. I feel this information may be important for the final interpretation of the results. Indeed, a much lower level of scPif1DN compared to scPif1 (under over-expressing conditions) would explain its reduced cellular toxicity under over-expressing conditions.

2) It would be interesting to check the length of the telomeres (by southern blot) after several cell cycles from the over-expression of scPif1 or scPif1DN. As previously reported, over-expression of scPif1 would induce a marked telomere shortening while it is not known if the same high levels of scPif1DN do the same.

Minor Points

1) Please check that the symbols (D, a and so on ...) are correctly reported in the final text.

2) lines 32-32. Please remove the two “and”.

3) line 54. Please remove “in this situation”.

4) line 64. Please remove “enabling the field to better understand” and add “leading to a better understanding of”.

5) line 71. Please remove italic format from the sentence “The other yeast strains used in this work are derived from”.

6) line 72. Please remove italic format from the sentence “which was a gift from Peter Walter”.

7) line 77. Please define “medium for protein overproduction” in detail.

8) Table 2 is “cut” in the final pdf document … please correct it.

8) Please add the symbol * to table 2 where necessary.

9) line 113. Please define if it is 1nM.

10) line 114. Please define how “the helicase loading strand was end-labelled” in detail.

11) line 115. Please define how the annealing of the “partially complementary oligonucleotides” was done in detail (i.e, buffer composition for DNA-DNA and DNA-RNA fork substrates).

12) line 120. Please define “binding buffer” in detail.

13) line 120. Please define “5x dye-free load buffer” in detail.

14) lines 197-198. Please clarify the sentence.

15) Figure 2: it would be helpful for not specialised readers to have a graphical view (scheme) of the telomere extension assay that shows how the substrate is (how it is labelled) and how it is elongated by the telomerase (in type I and II processivity). It would be also useful if the authors places Pif1 in these schemes either when it stimulates telomere elongation or when it inhibits it.

16) line 171. Should “25-125 nM” be replaced by 125-250nM in this sentence?

17) Also for Figure 4 I feel that graphical views of the biochemical reactions with a schematic representation of the substrates, their labelling and how Pif1 is supposed to act on them would help not specialised readers to better interpret the corresponding DNA binding and fork unwinding reaction and corresponding charts. These graphical views (schemes) could be placed beside each chart.

18) Line 321. In my view the majority would be more appropriate than “most”.

19) lines 351-359. I feel that this paragraph is not necessary, and it interrupts the flow of the discussion.

20) line 397. Please remove “and observations”.

Author Response

Our replies to your comments have been uploaded as a Word document (Reviewer 2.docx).

Reviewer 3 Report

To better understand the multitude of ways helicases contibute to genome maintenance, it is important to determine biochemical and biological functions of individual domains of PIF 1 and related helicases. In this manuscript, the authors compared a N-terminus truncated PIF1 protein with the full length version to gain an initial assessment of the role of N-terminus of PIF1 helicase of S. cerevisiae. The experiments are well designed and executed. The manuscript is well written and provide a novel indication of the separation of biochemical and biological functions of PIF-helicase via its N-terminus. Minor comments are below:

Recombinant PIF1 proteins are expressed in E. coli whereas Hrq1 is from baculovirus expression. Are there any post-translational modifications of PIF1, especially N-terminus, that might contribute to the differences observed in in vitro and in vivo behaviour?

Does the lack of growth defect in PIF1@N in Fig 3 suggest additional non-telomere related roles of PIF1?

As related to section 3.5 and related discussion, it is recommended that the authors test physical interaction between truncated PIF1 and hrq1, even if it is using the most traditional routine techniques.

Spell check for minor typos, e.g. lines 196-198; line 323.

Author Response

Our replies to your comments have been uploaded as a Word document (Reviewer 3.docx).
